# DESIGN AND EVALUATION FOR ROBUST CONTINUAL LEARNING

## ABSTRACT

Continual learning is the ability to learn from new experiences without forgetting previous experiences. Different continual learning methods are each motivated by their own interpretation of the continual learning scenario, resulting in a wide variety of experiment protocols, which hinders understanding and comparison of results. Existing works emphasize differences in accuracy without considering the effects of experimental settings. However, understanding the effects of experimental assumptions is the most crucial part of any evaluation, as the experimental protocol may supply implicit information. We propose six rules as a guideline for experimental design and execution to conduct robust continual learning evaluation for better understanding of the methods. Using these rules, we demonstrate the importance of experimental choices regarding the sequence of incoming data and the sequence of the task oracle. Even when task oracle-based methods are desired, the rules can guide experimental design to support better evaluation and understanding of the continual learning methods. Consistent application of these rules in evaluating continual learning methods makes explicit the effect and validity of many assumptions, thereby avoiding misleading conclusions.

## 1 INTRODUCTION

Continual learning is the ability to learn from new non-stationary data without *catastrophic forgetting* (McCloskey & Cohen, 1989; Goodfellow et al., 2013; Kemker et al., 2018) previously learned experiences (Thrun & Mitchell, 1995; Hsu et al., 2018; Parisi et al., 2019). Unlike *transfer learning* approaches which are focused on one-directional transfer from a source to a target task (Bengio, 2012; Yosinski et al., 2014) and are not concerned with forgetting the source task, continual learning requires the model to maintain acceptable performance for the source task while learning the new target task.

Various techniques have been developed to tackle catastrophic forgetting; however, the majority of the methods require a task oracle (identifier) to identify which task each input example belongs to (Kirkpatrick et al., 2017; Shin et al., 2017; Sener & Koltun, 2018). Often in experimental evaluations, a human plays the role of the oracle, with no intrinsic justification for the choice of tasks. Typical evaluations of continual learning methods generate synthetic tasks by splitting or permuting the input data and presenting the resulting tasks in sequence (van de Ven & Tolias, 2019; Aljundi et al., 2019a). For example, in SplitMNIST, a common dataset sequence used in continual learning, the ten digits are partitioned into chunks composed of two or more digits. Often, the data is split into five chunks, and each chunk has only two digits to classify. A task oracle (identifier) may label the chunks of the incoming data sequence as tasks. Specifically, we use *chunk* to avoid implicitly assuming these matches with task labels as this may provide unintentional information for performance boost. We may still choose a task oracle to have the same sequence but will need to be cognizant of the assumption. In Section 2, we propose six rules to help design experiments to test and understand these assumptions.

In general, a task oracle is not available, and a clear separation of tasks is unlikely to be defined or known (Caccia et al., 2020). Real-world applications are unlikely to exhibit strict sequencing of tasks; instead, they may exhibit mixing and transitioning between input distributions. It is important to be careful with assuming having a task oracle as the extra information provided will unfairly bias the evaluation of continual learning methods. Hence, in this work, we consider a variety of

input sequences that may occur in a continual learning problem. We demonstrate the more difficult scenario, which we called unrestricted scenario, where only limited task information is supplied to the methods during training.

Farquhar & Gal (2018) and van de Ven & Tolias (2019) suggest that continual learning research should shift away from task oracle-based methods. Although we share this sentiment, we do believe that a task oracle may still play a role in continual learning research, despite limiting real-world applications. However, researchers must apply robust experimental design to surface assumptions about learning scenarios.

In this work, we investigate the effects of task identification and experimental protocols in continual learning. Firstly, we propose six rules for robust evaluation of continual learning methods. Secondly, we extend published continual learning methods to support non-sequential tasks and evaluate the methods with a more extensive range of input arrival sequences and task partitionings to demonstrate the shortcoming of existing assumptions and protocols. We show a *sequential split* input sequence is most prone to catastrophic forgetting and current experimental protocols may not be evaluating the continual learning methods fairly. Finally, we discuss the application of the six rules on experimental design and interpretation of new experiments. We suggest the continual learning researchers should consider three different sequences: data, task identifier and evaluation identifier when designing experiments.

## 2 EXPERIMENTAL DESIGN FOR CONTINUAL LEARNING

Experimental and implementation choices have a significant effect on the performance of continual learning methods. While it is infeasible to consider all choices and their effects, it is still desirable to be able to fairly compare different methods and interpret the results. We propose six rules to aid in the design of robust experiments for continual learning.

### 2.1 RULE 1: TEST EXTREMES

Although it is infeasible to test all possible setups, experiments should cover extreme settings that can test the continual learning method to give a more expository understanding of the methods. When we allow different ordering of experimental sequences, there is an exponential number of possible sequences of data. In particular, the effects of the methods should compare to extreme settings. Continual learning experiments consider the sequential setting where the data are split into different sequences where each chunk of the split contains only a subset of classes. For example, splitting should be randomized rather than following some chosen order. In this case, we consider the extreme of the composition of the sequence, that is, grouping classes and their order of arrival to be random rather than alphabetical, numerical, or human categorization (cognitive or artificial). Typical experimental settings for continual learning consist of data splitting into clearly defined chunks where each contains only certain classes. The extreme in this case is the separation of tasks. Most continual learning experiments only consider the case where tasks are perfectly separated into chunks. Other separations in different extremes should also be considered, such as where the data is random, and all tasks may be present in every batch (i.e. the normal supervised learning setting). This extreme provides a baseline and ensures that the method does not rely on certain signals from the sequence's setup.

> **Rule 1:** *Identify and test extreme settings in experimental setup and parameters.*

### 2.2 RULE 2: USE CLEAR BASELINES

Results derived from experiments with a dependency on the data sequence can vary depending on how the data sequence is presented. Implementation or experimentation choices that seem arbitrary may, in fact, have a significant effect that is difficult to predict. Even when an analytical derivation may be possible, the work may be tedious and ad hoc. Hence, we propose including a clearly identified baseline equivalent to a 'random guess' for each experimental scenario.

In experimental protocols for continual learning, extra information may be provided to the methods such that random guess baseline performance with the same extra information may not match our

intuition. This means that providing a random guess classifier conforming to the experimental protocol is important to demonstrate the performance improvement attributable to the method.

A concrete example of this can be seen in the multi-head task scenario experiment used in many studies (von Oswald et al., 2020; Lopez-Paz et al., 2017; van de Ven & Tolias, 2019; Mirzadeh et al., 2020). For example, when splitting CIFAR100 into 10 tasks where each task has 10 classes, a task label is provided that limits the number of classes available to the classifier. One might naively think that random guess is 1% (1 out 100 classes), but due to experimental setup, the baseline is 10% (1 out 10 classes). Predicting random guess performance is difficult in more complicated experimental setups. Hence, providing a random guess classifier helps to identify the baseline of an experimental setup with ease.

> **Rule 2:** *Provide random guess implementations that conform to experimental setup for clear identification of baselines of the setting.*

## 2.3 RULE 3: TEST SIMPLE METHODS

Due to the extra information that naturally arises from the experimental setup in continual learning literature, it is important to provide simple methods that utilize this information. Many continual learning methods inherently uses task information which is usually not obvious. There is an attempt to move away from this paradigm towards a task-agnostic approach (Lee et al., 2020; Zeno et al., 2018; Aljundi et al., 2019a). However, regardless of the fairness or applicability of providing task information, when they are provided, other simple methods which also use this information should be compared to.

Naive multi-model, where each model is assigned a task, is a possible simple method to consider when task information is given. Each individual model is independent and there is no sharing between the models. The experiment supplies a task label for each input example, hence we can pick the model associated with the right task. This method uses the same information given to the continual learning methods to which we want to compare.

Naive multi-model may also be used in a task-agnostic setting such as using entropy from prediction as a proxy of task label or task change. All models will give an output for each example. The model with the lowest entropy output is used for classification for the example. Some continual learning researchers consider naive multi-model as the upper bound result that cannot be outperformed (Mirzadeh et al., 2020; Schwarz et al., 2018; Chaudhry et al., 2019; He et al., 2019; Titsias et al., 2020). However, we believe that this simply demonstrates that continual learning literature needs better experimental protocols. Naive multi-model uses the same information as task oracle based methods but is simpler and cannot make use of sharing to help combat forgetting and improve the learning of new data von Oswald et al. (2020), hence it should not be considered an upper bound. Some multi-task learning where tasks are considered as different objective functions (rather than as splitting data) do exhibit improvement through sharing Kendall et al. (2018). In this case, each input has multiple different objectives and using a shared model is better than multiple individual models.

> **Rule 3:** *Evaluate naive or simple methods for the experimental setting.*

## 2.4 RULE 4: RE-EVALUATE EXISTING BEHAVIOR

Often we assume that once some behavior of a model is established, it will continue to hold in further training and testing. However, this assumption is far from the empirical results. A clear example of this is continual learning itself where forgetting happens greatly (Kirkpatrick et al., 2017). The same data with the same training length and setup but with slightly different ordering can greatly affect the results of the models. This rule when applied to continual learning considers re-evaluating whether the previous task is learned. Continual learning experiments rarely test learning behavior by revisiting the same task. Hence, we propose to consider an experimental setup where task is recapitulated during training and how this may affect the proposed methods. One consequence of this rule is to require that methods can correctly identify data that they have previously seen, rather than considering all new data as a new task. In methods that rely on task labels, this is not an issue since task labels will be provided for the recapitulating data. However, for task-agnostic methods that must provide their own means to separate data, this becomes important. This test is useful in detecting

whether task-agnostic methods can reuse their previously learned components for new data of the same task or even previously seen data. Reusing existing components is essential, since the efficiency of training is better when we use generalized features that are already learned by the model. It also impacts the evaluation process, as in the task-agnostic setting, methods are greatly affected by the choice of task. For example, if there is a task-agnostic naive multi-model method then selecting the right model will significantly improve the prediction accuracy. Model size and memory usage will continuously grow if the method is unable to detect previously seen tasks. In the extreme, every new data may be considered a new task and the method may simply remember every example. This is a problem for many continual learning methods that rely on task boundary changes (Rebuffi et al., 2017; Kirkpatrick et al., 2017; von Oswald et al., 2020).

> **Rule 4:** *Perform experiments with settings that recapitulate incoming data.*

## 2.5 RULE 5: EVALUATE DEPENDENCY ON SEQUENCING

Experimental choices regarding data sequencing are often treated as arbitrary, when it is unknown whether they are in fact consequential. In online continual learning, one should evaluate experiments on different compositions of incoming data to ensure the choice of composition is inconsequential. Usually, the order of chunks will be shuffled and experiments repeat but one should also consider different possible formations of chunks. For example, one can consider different size chunks and completely random as pointed out in rule 1. Other possible effects of learning may be considered such as learning easier examples followed by hard examples. Or always have a proportion of hard examples mixed.

Other settings such as the scenario where each incoming data chunk consists of multiple different tasks rather than just one task should be evaluated. This setup will test whether the continual learning methods have a dependency on only allowing one task per chunk. Performance-wise, it will check the effect of extra information on the methods such that the methods will need to consider each chunk and their properties rather than just when they are changing to a new sequence of classes.

> **Rule 5:** *Identify and investigate different choices for the experimental sequencing.*

## 2.6 RULE 6: USE PROTOCOL-INDEPENDENT METRICS

Fundamental to continual learning is the ability to retain previously learned knowledge while obtaining new knowledge under constraints given. Hence, the evaluation of forgetting and sharing is important. Traditionally, metrics would be defined with the assumption that "tasks" are known beforehand, and a strict sequence of data is given. However, in many applications, tasks are either not known or ill-defined (Lee et al., 2020; Zeno et al., 2018; Aljundi et al., 2019a). Hence, only the final performance evaluation can be used, i.e., the final prediction accuracy for classification problems.

Examples of metrics that should be avoided or modified are forward and backward transfer, as these metrics assume data is split perfectly into tasks and arrives in a strict order (Lopez-Paz et al., 2017; Yoon et al., 2018; Schwarz et al., 2018; Chaudhry et al., 2018; Díaz-Rodríguez et al., 2018; Kemker et al., 2018). These metrics assume that data must come in sequence and one task followed by the other. Even in the most simple scenario of random data, these metrics break down, as just one example of a previously seen task will mean the metrics can not measure the difference in transfer.

Currently, the only unbiased method in identifying forgetting would be keeping track of each testing example (Toneva et al., 2019). Forgetting is defined to be misclassification after correct classification. It keeps track of examples and checks if they had ever been learnt (correctly classified). If, after more learning, the example fails to be correctly classified, then this is considered as forgetting. However, this is troublesome to perform and also difficult to give aggregate summaries. Hence, further work is required to find useful metrics for measuring continual learning.

> **Rule 6:** *Define performance metrics that derive from the data, rather than features of the experimental setup.*

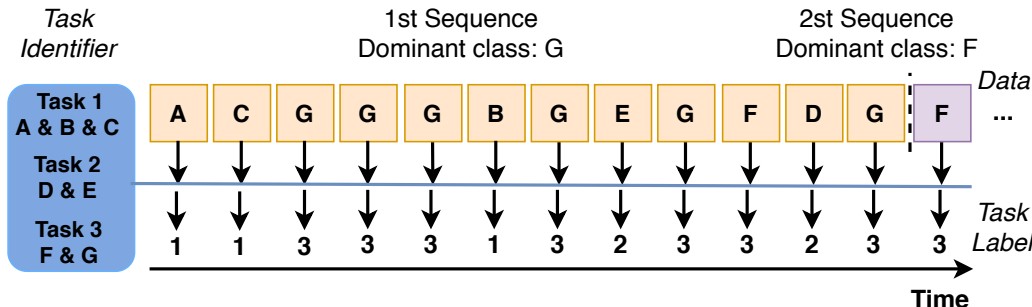

Figure 1: Example of dominant sequencing and dominant task identifier. The dominant class for the first sequence is class G ( 50%) while all other classes are present too. The second sequence have a different dominant class F. The task identifier has task 1 being larger than other tasks.

## 3 EXPERIMENTS

We tested different sequences and task identifiers on the MNIST and CIFAR100 datasets. The experiments focus on the more difficult online setting where data stream cannot be stored (Cui et al., 2016; Bottou, 1998) and the sequence in which examples are presented may affect learning. It is important to distinguish between the sequence in which data arrive and the sequence for task identifier, as a task oracle should not be assumed and the two sequences need not match.

### 3.1 SEQUENCE OF DATA ARRIVAL

In the online setting, each input example is only presented once. We consider a variety of different sequences of the input examples where each chunk $\mathcal{C} = (x_i, y_i)_{i=1}^N$ contains N examples:

**Random**: The data arrive in random order (rule 1). Results for random are in the appendix D.1.

**Split**: The input examples are split into a sequence of multiple chunks, $\mathcal{C}_1, \mathcal{C}_2, ...\mathcal{C}_n$, where each chunk consists of non-overlapping classes. For example, splitting MNIST into five chunks, where each chunk contains two classes (digits) (Yoon et al., 2018).

**Dominant**: Each split consists of a majority (55%) of examples from one dominant class, with the remainder of the split composed of examples from all other classes. This is an example of rule 5.

**SplitTwo**: The input examples are split into multiple chunks, $\mathcal{C}_1, \mathcal{C}_2, ...\mathcal{C}_{2n}$, where, $\mathcal{C}_1, ...\mathcal{C}_n$ have non-overlapping classes and, $\mathcal{C}_i$ and $\mathcal{C}_{n+i}$ have the same classes. This sequence is an application of rule 4 where grouping of classes is recapitulated. For our experiment, $\mathcal{C}_i \cap \mathcal{C}_{n+i} = \emptyset$.

### 3.2 TASK IDENTIFIER

Current task oracles commonly used in continual literature are unlikely to be given and may not be the best partitioning. To test this assumption, we evaluate different task separations. With the rules such as rule 1 and rule 5, we try to cover a range of identifiers. A task sequence is $\mathcal{I}_1, \mathcal{I}_2, ...\mathcal{I}_t$ where $I = \bigcup_i \mathcal{I}_i = \mathbb{N}_{\leq c}$ and $c$ is the number of classes. If the additional condition $\forall i, j, \mathcal{I}_i \cap \mathcal{I}_j = \emptyset$ when $i \neq j$ holds then it is called a partitioned task sequence. Given a partitioned task sequence, a perfect task identifier is $\forall x, f(x_{i,n}) = j$ if $i \in \mathcal{I}_j$ where $x_{i,n}$ is the nth example of class $i$.

**Split task identifier** Split task identifier (Sp=N), is a perfect task identifier where $\mathcal{I}_1, \mathcal{I}_2, ...\mathcal{I}_N$ is a partitioned task sequence of N tasks and $\forall i, |\mathcal{I}_i| = N/c$ For example, for the 10-class MNIST dataset, Sp=5 gives a task identifier which identifies five tasks each with two classes (digits). Note the task labels may not align with the data sequence, ie, the classes of $\mathcal{C}_i$ doesn't necessary equal $\mathcal{I}_i$. We only considered non-alignment in 4.4. Others assume the sequence for split identifier aligns with the data sequence. Further details and concrete example of sequences is given in appendix B.

**Dominant tasks identifier (Dom)**: This is also a perfect task identifier with the task sequence $\mathcal{I}_1, \mathcal{I}_2, \mathcal{I}_3$ where $|\mathcal{I}_1| = 0.2c$, $|\mathcal{I}_2| = 0.3c$ and $|\mathcal{I}_3| = 0.5c$

**Random task identifier (Rand)**: where examples are randomly allocated to a task label dynamically.

In the above list, only the random task identifiers have the ability to dynamically allocate tasks. All other identifiers have fixed task allocation, even in the online setting. We also included using entropy for model selection for multi-model. No restriction on prediction applies (Single head).

Figure 1 shows an example testing scenario where the data arrives in a dominant sequence fashion and is labelled by a dominant task identifier. Although the data sequence and task identifier are both dominant, their actual sequencing is different. This is different to traditional setting where all of the first sequences will be considered as one task. Here, it has three tasks with unequal frequency.

## 3.3 CONTINUAL LEARNING METHODS

We compared different continual learning models and baseline models using the task identifiers described in Section 3.2. The choice of methods attempts to cover the major types of continual learning algorithms reported in the literature. **Single model (Single)** provides a baseline method without continual learning as suggested in rule 3. **Elastic Weight Consolidation (EWC)** (Kirkpatrick et al., 2017) and **Synaptic Intelligence(SI)** (Zenke et al., 2017) are regularization methods. **Incremental Learner (iCaRL)** (Rebuffi et al., 2017) to cover incremental (Ristin et al., 2014) and example replay based methods. In inference, the model can be use for prediction or the best matching class from the exemplar set. The result shown is for the second case, exemplar set. Also, **Generative replay (GR)** (Shin et al., 2017) for another example of replay based methods. **Multi-Model (MM)**, where we have a model for each task is used as suggested in rule 3. **Dirichlet process mixutre model (NDPM)** (Lee et al., 2020) represents task-agnostic methods for CIFAR. Finally, we provide **random (rand)** and **random Multi-Model (randMM)** as suggested in rule 2 to provide random baselines that may not be obvious. The details of parameters used for the models and methods are in the supplementary materials.

Modifications are required on most methods as they are designed and optimized for split task sequences. Extension of the methods for non-sequential tasks is necessary for many scenarios, such as the combination of a dominant sequence of data with a split task identifier, as the dominant sequence is likely to include sequences of data which will be identified by the split task identifier as different tasks in each iteration. Hence, this is a further consideration that needs to be taken into account as most designs that rely on specific task knowledge will provide optimisation that does not apply to more general scenarios. For example, the iCaRL method is designed to incrementally add in examples of new classes as a fixed sequencing is assumed. As this assumption is removed, modifications are made so that the example set has to be recomputed at each batch for each observed class, which means the method is much more computationally expensive than assumed. Furthermore, originally a large number of examples are stored at the end of each task. This is removed in the online scenario and only examples of the current batch are used to compute which examples are stored. The tasks are not guaranteed to arrive in a specific order and interleave as seen in Figure 1.

## 4 RESULTS

In previous work, continual learning systems have been tested on task scenarios in which task labels perfectly matching the data sequence are given at both the training and testing stages. We also consider a more difficult setting where the task labels are not used to restrict prediction in this way, which we called unrestricted scenario. In this setting, task labels are only used to indicate task change for the continual learning method to activate. Whereas in the task scenario, task labels can be used to further restrict updates of the model and predictions to only relevant classes in both training and testing. The unrestricted scenario uses a single head, unlike the task scenario which uses multi-head for prediction. This means that the learning problem in the unrestricted scenario is more difficult as less information is assumed. Multi models in unrestricted setting still require selection of model in test time so we still provide task identifiers for model selection which is different to other methods. Hence, these results are separated by a line. The prediction for the selected model is still single head.

The results are the average over ten different seeds with the standard deviation indicated in brackets. More details of the model and parameters used for training can be found in the appendix A.1. The results for the random sequencing and offline (application of rule 1) are shown in the appendix, as accuracy for all methods are relatively high suggesting the problem setting may be too simplistic to meaningfully evaluate continual learning methods.

Table 1: Online Dominant MNIST. The result is average over 10 runs with std error in brackets.

| Method | Sp=1 | Sp=2 | Sp=5 | Dom | Rand | Entropy |
|---|---|---|---|---|---|---|
| Rand | 10.01(0.14) | 10.01(0.14) | 10.01(0.14) | 10.01(0.14) | 10.01(0.14) | N.A |
| Single | 93.84(0.24) | 93.84(0.24) | 93.84(0.24) | 93.84(0.24) | 93.84(0.24) | N.A |
| EWC | 93.84(0.24) | 93.47(0.35) | 93.47(0.35) | 94.76(0.24) | 93.47(0.35) | N.A |
| SI | 93.84(0.24) | 93.84(0.24) | 93.84(0.24) | 94.08(0.44) | 93.84(0.24) | N.A |
| GR | 93.85(0.31) | 91.35(0.24) | 91.35(0.24) | 91.35(0.24) | 91.35(0.24) | N.A |
| iCaRL w.e | N.A | 93.27(0.11) | 93.27(0.11) | 93.27(0.11) | 93.27(0.11) | N.A |
| MM | 91.69(0.29) | 95.81(0.23) | 98.73(0.13) | 96.84(0.40) | 87.93(0.39) | 91.47(0.34) |
| RandMM | 10.01(0.14) | 20.25(0.15) | 50.06(0.12) | 30.02(0.16) | 10.01(0.14) | 10.01(0.14) |

## 4.1 MNIST

Sequential partitionings are used in many evaluations of continual learning (Kirkpatrick et al., 2017; Shin et al., 2017; Rebuffi et al., 2017; Yoon et al., 2018). For example in splitMNIST, the data is split into five partitions of two different classes each. However, the task partitions are not necessarily aligned with the data sequencing. For instance, a two task identifier cannot align to a five-way partitioning of the incoming sequence.

In the unrestricted setting for splitMNIST, most regularisation results are around 20% which is a similar founding as Farqhuar Farquhar & Gal (2018) as their single head setting. Where as for multi-head these results are usually over 95%. However, in addition, we also consider various baselines and settings such as multi-model and random task identifiers to give a better understanding of the results. Due to the relative simplicity of splitMNIST we have left the full result and other discussions on individual results for the appendix D.2.

The results for the dominant sequence manage to retain high accuracy for all methods as shown in Table 1. Despite its reduction in examples for the non-dominant classes for each split, having a small number of examples for all classes in the dominant sequence avoided forgetting. Whereas in split sequences forgetting is more obvious as only classes of each split are present. Suggesting that if a small amount of data for different classes is present will combat forgetting significantly.

## 4.2 UNRESTRICTED AND TASK SCENARIO FOR SPLITCIFAR

Previous results show that the online split data is necessary to test the continual learning methods as other sequencing have a minimal reduction in performance. Hence, the split sequencing is also tested on the CIFAR-100 dataset as shown in Table 2. For this experiment, we consider 100 classes to be split into 10 chunks of 10 classes each. Similarly as before, the MM method exhibit advantage over other methods with split task identifier as each sub-model is only trained on a subset of the examples. Note NDPM do not use task information in unrestricted scenario.

Table 2: Online Split CIFAR-100 in unrestricted scenario

| Method | Sp=5 | Sp=10 | Sp=20 | Dom | Rand | Entropy |
|---|---|---|---|---|---|---|
| Rand | 0.97(0.01) | 0.97(0.01) | 0.97(0.01) | 0.97(0.01) | 0.97(0.01) | N.A |
| Single | 4.61(0.22) | 4.58(0.16) | 4.37(0.13) | 4.65(0.17) | 4.68(0.18) | N.A |
| EWC | 2.61(0.52) | 1.66(0.37) | 4.71(0.26) | 4.41(0.24) | 4.64(0.20) | N.A |
| SI | 4.27(0.27) | 4.65(0.25) | 4.43(0.19) | 4.48(0.24) | 4.39(0.24) | N.A |
| GR | 2.14(0.15) | 2.06(0.18) | 1.96(0.12) | 1.95(0.13) | 1.89(0.09) | N.A |
| iCaRL w.e | 7.76(0.15) | 7.68(0.26) | 8.05(0.21) | 8.07(0.12) | 7.93(0.23) | N.A |
| NDPM | — | 13.88(0.14) | — | — | — | — |
| MM | 10.49(0.26) | 18.99(0.40) | 31.41(0.40) | 4.28(0.17) | 1.10(0.02) | 1.95(0.09) |
| RandMM | 4.97(0.04) | 10.00(0.07) | 19.98(0.11) | 3.07(0.04) | 0.97(0.01) | 0.97(0.01) |

The same experiment of split CIFAR was also run in the task scenario to demonstrate that the task scenario may not be a useful protocol for evaluation. The single baseline and regularization methods in the unrestricted scenario will only achieve 1-4% whereas in the task scenario the performance increased to 5-30% in appendix E. Perfect task identifiers will be able to provide information that can be used to reduce the complexity of the problem. Split task identifier has better performance on multi-model as it is able to train only on relevant output nodes matching classes of arriving data chunk even in the single-head setting. Further analysis of this matching assumption is explored in 4.4.

### 4.3 SPLITTWOCIFAR

We study recapitulating data (rule 4) to test the effect of previously seen classes reappearing. The results shown in Table 3 are similar to the original splitCIFAR but with slightly lower performance. This is likely due to the training data been further spread out hence more forgetting will occur. For sp=5, EWC seems to have improvement likely due to slightly more regularization (from 5 times to 10). This result suggests that methods that claim to only need weaker assumptions for task oracle in form of task change (activate method when task change detect) is usually making more assumptions than just task change if recapitulating data is not tested.

Table 3: Online SplitTwo CIFAR-100 in unrestricted scenario

| Method | Sp=5 | Sp=10 | Sp=20 | Dom | Rand | Entropy |
|---|---|---|---|---|---|---|
| Rand | 1.01(0.02) | 1.01(0.02) | 1.01(0.02) | 1.01(0.02) | 1.01(0.02) | N.A |
| Single | 4.25(0.27) | 4.34(0.21) | 4.29(0.22) | 4.31(0.22) | 4.31(0.16) | N.A |
| EWC | 3.21(0.46) | 1.28(0.09) | 4.43(0.18) | 4.45(0.17) | 4.29(0.21) | N.A |
| SI | 4.31(0.23) | 4.28(0.24) | 4.34(0.27) | 4.31(0.22) | 4.15(0.19) | N.A |
| GR | 1.87(0.18) | 2.34(0.18) | 2.08(0.11) | 2.19(0.18) | 1.98(0.17) | N.A |
| iCaRL w.e | 7.74(0.24) | 8.15(0.19) | 7.88(0.25) | 7.83(0.20) | 7.87(0.19) | N.A |
| NDPM | — | 13.85(0.07) | — | — | — | — |
| MM | 9.96(0.36) | 17.63(0.25) | 30.76(0.53) | 4.42(0.20) | 1.02(0.05) | 1.51(0.09) |
| RandMM | 5.02(0.07) | 9.89(0.08) | 19.89(0.10) | 3.00(0.04) | 1.01(0.02) | 1.01(0.02) |

Table 4: Split CIFAR-100 different training and evaluation sequence in task scenario

| Method | es | es-ed | ea | td | td es | td es-ed |
|---|---|---|---|---|---|---|
| Random | 10.00(0.07) | 1.52(0.04) | 0.97(0.01) | 1.52(0.04) | 10.00(0.07) | 1.49(0.04) |
| Single | 31.36(1.22) | 8.62(0.42) | 6.21(0.34) | 8.13(0.54) | 30.20(1.79) | 8.00(0.43) |
| EWC | 27.69(2.99) | 6.81(1.12) | 4.72(0.90) | 8.31(0.49) | 30.51(1.46) | 8.72(0.37) |
| SI | 30.53(1.08) | 8.38(0.28) | 6.31(0.41) | 8.31(0.56) | 30.86(1.16) | 8.33(0.43) |
| GR | 11.89(0.16) | 2.44(0.22) | 1.69(0.06) | 2.32(0.13) | 11.90(0.12) | 2.46(0.09) |
| iCaRL w.e | 32.79(0.51) | 10.98(0.35) | 7.64(0.22) | 10.61(0.20) | 33.18(0.48) | 10.75(0.18) |
| MM | 18.29(0.32) | 16.16(0.44) | 15.30(0.24) | 11.69(0.33) | **71.70(0.99)** | 18.02(0.86) |
| RandomMM | 10.00(0.07) | 10.00(0.07) | 10.00(0.07) | 9.95(0.11) | 66.09(1.11) | 15.25(0.36) |

### 4.4 THREE SEQUENCES IN CONTINUAL LEARNING

In this section, we consider split CIFAR-100 in the task scenario using the $sp = 10$ task identifier with various experimental settings as suggested by rule 5. We show that there is three different partitioned sequences to consider in continual learning: 1)Data, $sq_1 = (\mathcal{D}_i)_{i=1}^n$ where $\mathcal{D}_i$ contain the classes encounter in $\mathcal{C}_i$. 2)Task identifier sequence, $sq_2 = (\mathcal{I}_i)_{i=1}^t$, sequence of task identifier for supplying task information to continual learning methods. 3)Evaluation identifier sequence, $sq_3 = (\mathcal{J}_i)_{i=1}^e$ for restricting prediction. Each term of the sequences is of the same size for their respectively sequence and $n = t = e = 10$ unless specified otherwise. The evaluation identifier is used to identify which tasks are seen in each batch and hence restrict prediction to all the possible classes cover by the tasks. Table 4 shows the result of different task and evaluation identifiers.

For evaluate all (*ea*), the data and task identifier have the same sequence and $e = 1$. However, there is no restriction for prediction to emulate single head setting $sq_1 = sq_2 \neq sq_3$. We have $\mathcal{D}_i = \mathcal{I}_i$ for all $i$ and $sq_3 = \mathcal{J}_1 = \mathbb{N}_{\leq 100}$. This result shows similar performance to sp=10 in Table 2 suggesting that multi-head (evaluation identifier) provide most of the benefit over task identifier.

For evaluate split(*es*), the evaluation identifier is a split10 task identifier and default to have the same sequence as data, $sq_1 = sq_2 = sq_3$, $\mathcal{D}_i = \mathcal{I}_i = \mathcal{J}_i$ for all $i$. Hence, this setting is equivalent to $sp = 10$ in the task scenario. We consider evaluate split different(*es-ed*), where $sq_1 = sq_2 \neq sq_3$, $\mathcal{D}_i = \mathcal{I}_i \neq \mathcal{J}_i$ for all $i$. different to ea, the evaluation identifier still provide prefect matching every 10 classes to 1 task. However, as these classes mismatch with data there is less restriction hence poor performance. Conceptually, this is a partial multi-head setting. This shows despite having two perfect task oracles there is still more assumption being made for the single task oracle which aligns with data. Similar behavior can be seen in task identifier differ (*td*), where $sq_1 \neq sq_2 = sq_3$.

Interestingly, for *td-es*, where $sq_1 = sq_3 \neq sq_2$, performance for multi-model is the overall highest at 71.7%. Despite, $sq_2$, having a mismatch with the data there is a large performance improvement as evaluation is in sync with the data ($\mathcal{D}_i = \mathcal{J}_i \neq \mathcal{I}_i$ for all $i$). This is because the chosen model will be trained on other mismatch classes, hence, the restriction will further focus on matched classes which can be observed from the high randMM result of 66.1%. Note, multi-model in task scenario used the exact same information as all other methods for this setting. This again demonstrates how evaluation identifier is what is providing most of the performance gain. As in *td es-ed*, $sq_1 \neq sq_2 \neq sq_3$, we fall back to similar performance as *td*. Concrete example of sequences is shown in appendix B.

Our analysis on task oracle and single vs multi-head differs from existing work Farquhar & Gal (2018); van de Ven & Tolias (2019); Caccia et al. (2020) in two ways. First, we show there is more assumption on task oracle than a perfect task oracle. We show that most task oracle based methods not only assume a perfect task identifier which maps examples to task perfectly but there is also an additional assumption that the group of classes perfectly aligns with the incoming data. We believe most of the assumption is unrealistic as in most cases where the assumption is made the experiment could simply randomize the data in training rather than a sequential presentation. Second, we give a new perspective by considering three different sequences and also provide various baselines for different settings to provide a more robust and complete understanding of the methods in cases where the assumptions are desired and considered appropriate.

## 5 DISCUSSION

Designing robust experimental protocol is difficult due to the wide range of possible variations and small implementation details that may greatly affect results. In this paper, we proposed six rules in an attempt to cover many important experiment aspects to consider for continual learning. We demonstrated that the assumptions and testing protocol used in continual learning are unable to accurately measure the effectiveness of previously proposed continual learning methods. Many of the published performance results appear to be artifacts of the experimental protocol.

We consider a wide range of data sequences and task identifiers using rule 1, 4 and 5 to demonstrate various assumptions existing works made and the results without these assumptions. These experiments also lead to observations such as having a little bit of data from all classes mitigate forgetting as shown in dominant MNIST. The availability of task labels for testing simplifies the learning problem significantly in task oracle base methods. This is further shown in our analysis on the effect of task oracle by considering three different sequences: data, task and evaluation. The flexibility of implementations improves when reducing assumptions by removing optimization that rarely holds. With the application of the rules, we show various baselines that are useful to include for a better understanding of what information the experiments are providing to the methods. For example, multi-model outperforms all other continual learning methods as it can reduce the original problem to a simpler problem while using the same amount of information from the task oracle. Hence, the choice of testing protocol and settings is essential for comparing continual learning methods fairly.

As shown in the results, various random and random multi-model(rule 2) results have different values to what one might intuitively guess. Hence, providing these baselines helps identifying which information is assumed and what the baseline results are. Recapitulating tasks (rule 4) are especially important for task-agnostic methods (Aljundi et al., 2019b; Zeno et al., 2018; Rao et al., 2019), which to the authors best knowledge has only been considered in OSAKA (Caccia et al., 2020). The work considers task transition to be model by a Markov chain and uses $\alpha$ parameter to decide how likely the next data will be drawn from the current task. Hence, recapitulation occurs when it transitions back to previously visited task. However, OSAKA did not consider an extreme scenario as suggested by rule 1 by considering very low $\alpha$ values where transitions between tasks are prevalent.

Previously proposed metrics, such as forward transfer, and backward transfer, require a task oracle and sequential tasks. However, in task-agnostic scenarios, the sequence of data cannot be assumed and task labels is not be available. A good example of applying rule 6 where new metrics only depends on data can be seen in the work by Toneva et al. (2019), where forgetting is measured by tracking if an example is incorrectly identified after it was correctly identified. In future work, new metrics can be proposed to more efficiently aggregate forgetting than Toneva et al. (2019). We believe the rules proposed provide schemes to identify useful settings and baselines to consider and perform which produce a more robust and complete investigation of continual learning methods.

## REPRODUCIBILITY STATEMENT

Our work consider various experimental settings for continual learning. The range of data sequences is given in Section 3.1 and task identifiers in Section 3.2. Different continual learning methods are explained in Section 3.3. Their parameters and settings are supplied in the appendix A. We briefly cover implementation considerations in Section 3.3 for continual learning methods to be applicable in a more general setting. In Section 4.4, we have cover the three main type of sequences a continual learning experiment should consider. We have explained settings where the sequence is different and the case where they are the same. Concrete example of the sequences are also given in appendix B.

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

# A   CONTINUAL LEARNING MODEL PARAMETERS

- **Single model (Single)**: a baseline with 3 layers for MNIST and resnet for CIFAR without continual learning method applied.
- **Elastic Weight Consolidation (EWC)**: also a regularization method which uses the Fisher information between tasks to reduce catastrophic forgetting (Kirkpatrick et al., 2017).
- **Synaptic Intelligence(SI)**: a regularization method based on the running sum of the gradient for different tasks. This additional loss is added to a single model.
- **incremental Learner (iCaRL)**: stores examples for each task and uses them to classify new data. (Rebuffi et al., 2017).
- **Generative replay (GR)**: trains both a standard single-model classifier and a generator. For each task, the single model is trained on new data and pseudo-data generated by the generator from previously encountered data (Shin et al., 2017).
- **Multi-Model (MM)**: a number of smaller, separate models in which the total number of model parameters is the same as that of a single model.
- **Neural Dirichlet Process Mixture Model (NDPM)**: a task-agnostic continual learning method which uses Dirichlet process mixture model to combat forgetting (Lee et al., 2020).

## A.1   TRAINING PARAMETER

In all experiment, the batch size of 128 is used. Each experiments is run with 10 different seeds so that average result is reported.

### A.1.1   MNIST

The number of iteration used for MNIST in the off-line setting is 2000 which translate to about 4 epoch for MNIST dataset. Learning rate is set at 0.001 and Adam optimizer is used.

In online randomMNIST sequence, the number of iterations used in each split is 470. This will translate to about 1 epoch.

In domMNIST sequence, the number of iterations used in each split is 47 ( 6k example). This will translate to about 1 epoch so that data are only seen once.

In splitMNIST sequence, the number of iterations used in each split is 97( 12k example). This will translate to about 1 epoch as there are 5 splits.

### A.1.2   CIFAR

The learning rate is set at 0.01 and SGD optimizer is used for CIFAR experiments. In splitCIFAR sequence, the number of iterations used in each split is 40 ( 5k example). This will translate to about 1 epoch as there are 10 splits. In splitTwoCIFAR sequence, the number of iterations used in each split is 20 ( 2.5k example). This will translate to about 1 epoch as there are 10 splits.

## A.2   SINGLE MODEL

The single model is a 3 FC layer neural network with 400 neurons for MNIST for each of the middle layers before the output layer. ReLu is used as the activation function. In the CIFAR experiment, the same resnet architecture with Lee et al. (2020) is used.

## A.3   SYNAPTIC INTELLIGENCE

The regularization strength is c=0.1.

## A.4   ELASTIC WEIGHT CONSOLIDATION

The regularization strength is $\lambda = 5000$ for MNIST and $\lambda = 10$ for CIFAR.

## A.5 ICARL

The budget is 500 which is about 1% of the MNIST and CIFAR100 dataset. Exemplar is added to training in each batch.

## A.6 GENERATIVE REPLAY

The encoder and decoder of the variational autoencoder model share the same parameter with the single model with the latent dimension being 100. The optimizer for the VAE is ADAM optimizer.

## A.7 NAIVE MULTI MODEL

This method has multi small model with the aggregate size is the same as the single model baseline. In the MNIST experiments, the dynamic task labeller, som and rand are limited to having 15 models with hidden layer size of 30. Other static task labellers will have the same number of smaller models as the number of tasks identified. Specifically, the number of models, 5,3,2 correspond to hidden layer size of 90, 146, 214 respectively. For CIFAR experiments, the random task identifier is limited 10 models. The number of models 10,5,3 correspond to number of filter of 20, 28, 36 respectively. In the case where there is more task identified than models, the last model will be used for all other models.

## A.8 NEURAL DIRICHLET PROCESS MIXTURE MODEL (NDPM)

We use the setting provided by Lee et al. (2020) in cifar100-cndpm with a modification of short term memory (STM) to 500 to match the bandwidth with other replay based methods.

# B SEQUENCE EXAMPLES

In this section, we give concrete examples of sequence on MNIST. Suppose we have **split** MNIST where the data is split into 5 chunks, $\mathcal{C}_1, \mathcal{C}_2, ...\mathcal{C}_5$. The classes encounter in each chunk are $\mathcal{D}_1 = \{3, 9\}, \mathcal{D}_2 = \{1, 2\}, \mathcal{D}_3 = \{4, 0\}, \mathcal{D}_3 = \{5, 3\}, \mathcal{D}_5 = \{6, 7\}$ respectively. This is the *data sequence*, $sq_1$. Note that the classes encounter is not in numerical order which is a common assumption.

When we assume data and task sequence align the order above will be used for the split identifier. For example, $Sp = 2$ split task identifier will have the task identifier sequence, $sq_2$, of $\mathcal{I}_1 = \{3, 9, 1, 2, 4\}, \mathcal{I}_2 = \{0, 5, 3, 6, 7\}$. $\mathcal{I}_1$ contain classes of chunk 1,2 and half of 3 (the left over) and $\mathcal{I}_2$ the remaining. Similar, $Sp = 5$ split task identifier will have $\mathcal{I}_i = \mathcal{D}_i$ for all i. When the data and task sequence do not align (**td** case in Section 4.4) then $\exists i$ s.t $\mathcal{I}_i \neq \mathcal{D}_i$. It is possible to have $Sp = N$ where $N > 5$. Lets consider the case when $N = 7$, we will have 4 classes that is perfectly identified and the remaining 6 classes splits into 3 sets.

An example of dominant task identifier sequence, can be $\mathcal{I}_1 = \{2, 9, 4, 7, 8\}, \mathcal{I}_2 = \{0, 1, 6\}, \mathcal{I}_3 = \{3, 5\}$. This sequence is clearly different to $sq_1$ as the number of term do not match.

When we have the same identifier for task identifier and evaluation identifier then $sq_2 = sq_3$ and the identifier is of the same type. We could consider a case where we use dominant task identifier (as above $sq_2$ is $\mathcal{I}_1 = \{2, 9, 4, 7, 8\}, \mathcal{I}_2 = \{0, 1, 6\}, \mathcal{I}_3 = \{3, 5\}$), for training and split task identifier ($sq_3$ is $\mathcal{J}_1 = \{3, 9, 1, 2, 4\}, \mathcal{J}_2 = \{0, 5, 3, 6, 7\}$) for evaluation restriction then $sq_2 \neq sq_3$.

In the case where all three sequences is of split type and equal number of split the sequence can still not match. This is the same setting as *td es-ed* in Section 4.4. Suppose data is split into 5 chunks as above $sq_1$ is $\mathcal{D}_1 = \{3, 9\}, \mathcal{D}_2 = \{1, 2\}, \mathcal{D}_3 = \{4, 0\}, \mathcal{D}_3 = \{5, 3\}, \mathcal{D}_5 = \{6, 7\}$. The task identifier is of type $sp = 5$ with $sq_2$ is $\mathcal{I}_1 = \{2, 4\}, \mathcal{I}_2 = \{1, 8\}, \mathcal{I}_3 = \{3, 0\}, \mathcal{I}_3 = \{5, 7\}, \mathcal{I}_5 = \{6, 9\}$. The eval identifier is also of type $sp = 5$ but with $sq_3$ is $\mathcal{J}_1 = \{2, 9\}, \mathcal{J}_2 = \{1, 7\}, \mathcal{J}_3 = \{3, 8\}, \mathcal{J}_3 = \{5, 0\}, \mathcal{J}_5 = \{6, 4\}$. In this case, $sq_1 \neq sq_2 \neq sq_3$

## C  OFFLINE SETTING

In offline setting, all of the past data can be used for training and evaluation. In this setting, the order of how data arriving is not required as all data are assumed to be available. An application of rule 1 where randomness of data is at its extreme. In the offline setting only random sequence of the data will be considered as the data are all available at the start of the training procedure. Multiple epoch is allowed in the offline setting. The result is shown in table 5.

Table 5: Offline Random MNIST

| Method | Sp=1 | Sp=2 | Sp=5 | Dom | Rand |
|---|---|---|---|---|---|
| Single | — | — | 97.76(0.08) | — | — |
| SI | 97.76(0.08) | 97.76(0.08) | 97.76(0.08) | 97.76(0.08) | 97.74(0.08) |
| EWC | 97.76(0.08) | 97.86(0.05) | 97.86(0.05) | 97.86(0.05) | 97.86(0.05) |
| iCarL using exe | 96.78(0.04) | 96.78(0.04) | 96.78(0.04) | 96.73(0.05) | 96.78(0.04) |
| GR | 97.83(0.08) | 97.75(0.04) | 97.75(0.04) | 97.75(0.04) | 97.75(0.04) |
| MM | 97.76(0.08) | 98.14(0.07) | 99.40(0.07) | 98.17(0.10) | 90.19(0.10) |

Random sequence of MNIST with different task identifier and continual learning method all perform similarly with reasonable accuracy. There is only a single result for the single model as it does not use task information for learning. When the inputs arrive in random order different tasks are encountered which trigger continual learning methods. All continual learning methods will activate regularly with task change detected. Moreover, as inputs order are random, the inputs for different classes are encounter throughout the training so the effect of forgetting is diminished. The result for the multi-model method is the best for $sp = 2$, $sp = 5$ and dom as a perfect class to task mapping is provided thus perfectly segregating subset of classes on to different model which reduces the difficulty. On the other hand *rand* does not have this advantage since they can have examples from a single class map to different tasks.

## D  ONLINE MNIST

### D.1  ONLINE RANDOM MNIST

The result for the random sequencing is shown in table 6. There is a slight decrease in the accuracy likely due to the reduction in the number of iteration used to train the models. Similarly to the offline case the result, for a particular method, the result does not change even with different task identifier. The base single model also seems to perform minimally better likely due to less overfitting compare to other the exemplar and replay methods. Note that in online random setting, iCarL will not construct exemplar set as there is only one task. Hence, the result is omitted.

### D.2  SPLITMNIST IN UNRESTRICTED SCENARIO

The MNIST result is tested on a unrestricted scenario where task information is provided in the training stage only for the continual learning model and is not used in prediction. The dataset has 60,000 training examples and 10,000 for testing. Different task identifiers all exhibit similar results for the regularization based continual learning methods.

Table 6: Online Random MNIST

| Method | Sp=1 | Sp=2 | Sp=5 | Dom | Rand |
|---|---|---|---|---|---|
| Single | 96.40(0.07) | 96.40(0.07) | 96.40(0.07) | 96.40(0.07) | 96.40(0.07) |
| SI | 96.40(0.07) | 96.40(0.07) | 96.40(0.07) | 96.40(0.07) | 96.40(0.07) |
| EWC | 96.40(0.07) | 96.44(0.08) | 96.44(0.08) | 96.44(0.08) | 96.44(0.08) |
| iCarL using exe | — | 94.35(0.10) | 94.35(0.10) | 94.35(0.10) | 94.35(0.10) |
| GR | 96.22(0.08) | 96.22(0.09) | 96.22(0.09) | 96.22(0.09) | 96.22(0.09) |
| MM | 94.94(0.22) | 96.88(0.20) | 99.15(0.08) | 97.78(0.16) | 87.26(0.16) |

The results drop catastrophically in this scenario as shown in Table 7. The accuracy is similar to only predicting correctly on one split ( 20%) and completely inaccurate prediction on other splits. This suggests the models are over-fitting at the final split and the continual learning methods are unable to mitigate this. The iCaRL method has better performance than other methods as it exemplars are replayed during training and so it will include a small number of previous classes to combat forgetting. The exemplar set has the highest accuracy which indicates that the models are forgetting drastically. Generative replay achieved 40.6% for $sp = 5$ task identifier which perfectly align with the incoming data. This shows assuming task identifier may give extra information that is not attended. This is an example of rule 1 (testing extreme) and 5 (different choices) where consider various task identifiers which give us more insights into the methods. The high accuracy for MM is due to the allocation of data onto the respective model for each task thus for each sub-model only the example the class is trained on it. Similarly, for $sp = 5$ task identifier with MM, rule 3, (simple method) also has the highest performance of 99.13%. According to rule 2 (random baseline), we provided a random baseline which is also the highest of 50%. Other task identifiers do not have the same random or MM performance suggesting again that $sp = 5$ provides extra information. On the other hand, interestingly having weaker task identifier such as Sp=2 or even random still retain or outperform sp=5 (perfect match with data sequence) for the case of iCaRL and regularisation. Hence, suggesting the normally picked task oracle sp=5 may not be the best.

### D.3 DOMINATE MNIST

The regularization methods are unaffected by which task identifier as the methods used task change event to activate regularization. Task change will occur often since each dominant sequence contains all task. Generative replay method in the dominant sequencing shows varying results. It should be noted that in the one task setting generative replay is not activated as no tasks change will occur. For naive multi-model continual learning, SOM task identifier exhibits almost 3% better accuracy than random task identifier.

## E EXPERIMENTS IN TASK SCENARIO

We performed this on in the splitCIFAR experiment where the data is split into 10 splits with 10 classes each. The dataset is split into 50,000 training examples and 10,000 for testing.

Table 7: Online Split MNIST in unrestricted scenario

| Method | Sp=1 | Sp=2 | Sp=5 | Dom | Rand |
|---|---|---|---|---|---|
| Rand | 10.16(0.11) | 10.16(0.11) | 10.16(0.11) | 10.16(0.11) | 10.16(0.11) |
| Single | 19.31(0.29) | 19.31(0.29) | 19.31(0.29) | 19.31(0.29) | 19.31(0.29) |
| EWC | 19.31(0.29) | 19.51(0.93) | 19.68(0.08) | 19.01(0.54) | 19.54(0.09) |
| SI | 19.31(0.29) | 19.31(0.29) | 19.31(0.29) | 19.33(0.28) | 19.31(0.29) |
| GR | 19.53(0.10) | 24.28(1.52) | 40.59(1.52) | 23.71(2.48) | 19.99(0.38) |
| iCaRL w.e | 24.28(1.15) | 66.87(1.44) | 69.17(0.98) | 69.78(0.87) | 70.75(0.99) |
| MM | 17.85(1.31) | 30.63(0.48) | **99.13(0.06)** | 40.14(3.49) | 19.98(0.20) |
| RandMM | 10.16(0.11) | 20.07(0.08) | 49.86(0.12) | 30.17(0.13) | 10.16(0.11) |

Table 8: Online Split CIFAR-100 in task scenario

| Method | Sp=5 | Sp=10 | Sp=20 | Dom | Rand | Entropy |
|---|---|---|---|---|---|---|
| Rand | 4.97(0.04) | 10.00(0.07) | 10.00(0.07) | 1.00(0.02) | 0.97(0.01) | N.A |
| Single | 17.49(1.32) | 32.47(0.78) | 30.33(1.23) | 6.82(0.32) | 6.46(0.38) | N.A |
| EWC | 18.11(2.10) | 28.92(2.54) | 27.20(2.12) | 7.15(0.31) | 6.57(0.37) | N.A |
| SI | 18.50(1.09) | 27.92(2.28) | 30.48(1.44) | 6.48(0.35) | 6.69(0.30) | N.A |
| GR | 6.85(0.19) | 11.89(0.14) | 11.91(0.12) | 1.82(0.11) | 1.80(0.12) | N.A |
| iCaRL w.e | 22.28(0.52) | 32.69(0.52) | 33.85(0.28) | 8.07(0.23) | 7.71(0.18) | N.A |
| NDPM | **30.87(0.30)** | **42.05(0.44)** | **42.33(0.40)** | **13.80(0.16)** | **13.81(0.10)** | N.A |
| MM | 10.56(0.20) | 17.94(0.45) | 31.75(0.45) | 3.93(0.18) | 1.10(0.05) | 1.69(0.15) |
| RandMM | 4.97(0.04) | 10.00(0.07) | 19.98(0.11) | 3.07(0.04) | 0.97(0.01) | 0.97(0.01) |

Table 9: Online domain permMNIST

| Method | Pdom |
|---|---|
| Single | 63.04(0.98) |
| SI | 63.04(0.98) |
| EWC | 77.41(0.92) |
| iCaRL using exe | 79.84(0.54) |
| GR | 37.4 (2.26) |

The results is shown in Table 8. We can observe much higher performance for split task identifiers for all methods compare to unrestricted scenario. NDPM with the restriction during prediction also give a large performance boost compare to Table 2 in the unrestricted scenario.

## F    EXPERIMENT ON DOMAIN

This section contains the result of testing a domain continual learning scenario where every tasks is to predict the digits of 0 to 9 but each domain is a different permutation. The perfect domain identifier will always identify the domain of input correctly.

Unrestricted and domain scenario will always have all classes used in training, prediction or evaluation. The main difference of the domain scenario is that the domain between tasks is changing. Every task should have the same outputs but in a different domain.

Task scenario will only use classes seen in current tasks. For training prediction, the task scenario will have a list of active classes which each element of the list contains only the classes seen at each task. In evaluation, the scenario will have classes of present tasks in the current batch as active classes.

