# OpenReview forum: "Design and Evaluation for Robust Continual Learning"
_ICLR.cc/2022/Conference — ICLR 2022 Submitted_

### Official Review · Reviewer_rfDE · 2021-10-24

**Correctness:** 4
**Technical Novelty And Significance:** 1
**Empirical Novelty And Significance:** 2
**Recommendation:** 5
**Confidence:** 4

**Main Review:**

The continual learning literature uses a proliferation of different benchmark settings and evaluation protocols, and it’s often difficult to determine which of these a paper is using without delving into code or reading appendices or even having to email the authors. As the authors of this paper state, these choices can have a dramatic impact on results. So efforts to tidy up this methodolical confusion and to set out best practice, such as the present paper, are very much needed. However, papers of this sort are, in my view, not appropriate for a research-focused conference such as ICLR, as they are purely methodological and don’t offer new techniques or results or analysis. The paper reads more like a thesis chapter than a research paper. It might be better suited as part of a longer journal paper whose main contribution is a new continual learning method. Or perhaps an arXiv paper. The area chair may disagree, of course, as (I’m sure) will the authors. Notwithstanding its suitability for an ML conference, I also feel the paper covers ground that has already been covered by Hsu et al (2018) and van de Ven & Tolias (2019). (Both are cited in this paper. Notably, these are both arXiv-only papers, as far as I know.) Although the present paper offers some further thoughts and some useful new benchmark figures, I don’t feel this is enough novelty to warrant an ICLR paper.

The standard of English is poor. Mostly this isn’t a barrier to understanding the paper (and it doesn’t impact my score). However, there are a few sentences where I cannot even guess the intended meaning. Here are some examples:

p.5, fig.1, caption: “Contrasting, the task identifier have task 1 being larger than other tasks”

p.5, last para.: “We also include entropy for model selection for multi-model and does not have any restriction on prediction”

p.6: “In Section 4, the second case is used and labelled iCaRL using exe.”

p.9: “Reducing assumptions improves the flexibility of the implementations as it remove optimization that rarely holds.”

**Summary Of The Paper:**

This paper a) attempts to clarify the methodological assumptions that underlie the experimental results of many papers in the continual learning field, b) proposes a number of rules for benchmark settings and evaluation protocols in order to show the true capabilities of continual learning methods under challenging conditions, and c) benchmarks a number of existing methods following the guidelines in (a) and (b).

**Summary Of The Review:**

The paper is a laudable attempt to improve the methodology of continual learning research. But a lot of the ground it covers has been covered in other papers. And it’s more like a thesis chapter / journal paper section than a standalone contribution to the field.

---

> ### Author Response · Authors · 2021-11-22
> **Response to rfDE**
>
> We thanks the reviewer for their comments and understanding of the importance of methodology for evaluation. Following is our response to address their concerns:
>
> **Q1 Appropriateness for ICLR and purely methodological:**
>
> We believe our work is appropriate for the conference as continual learning is an empirical research field hence without robust evaluation the results of many works are insubstantial and inconclusive.
> As mentioned by the reviewers, the sensitivity and difficulty of the evaluation make the research field convoluted and unreliable.
> For instance, many recent task-oracle based works still failed to beat the naive multi-model baseline as mentioned in section 2.3.
> We think we have provided interesting and useful results and analysis such as the separation of task oracle from data sequence and model and evaluation, different task oracle and, new data sequences.
>
> A key contribution and message of our paper are to give a useful set of rules that all should be followed when experimenting in continual learning (or more general ML settings).
> Continual learning methods' abilities are hard to grasp due to the vast and messy settings.
> Despite some rules being followed in some work, most works still lack adherence to multiple rules.
> The rules also are for moving towards a more online and sequential base framework by reducing the bias of experimental design orientating with the task.
>
> **Q2 Novelty compares with related works:**
>
> We think our work provides substantial additional contribution for experimentation and is also different to existing related works.
> We provide a set of rules to act as a guideline for a better protocol for the continual learning community.
> Our framework is more general and the analyses cover more difficult and diverse settings.
> We hope to convince the community to remove implicit assumptions or make them clear.
> We wish the reviewers clarify "Notably, these are both arXiv-only papers, as far as I know."
> We are unsure of the meaning implied. Is it that arXiv results can be disregarded as they are not formally published or if arXiv results are important.
> We think Hsu et al (2018) and van de Ven & Tolias (2019) are highly important works.
> The papers are deserving of publication as despite being arXiv both papers are highly cited and many citations are from a recent publication in top venues.
>
> Following we address the difference of our work with existing work in detail.
>
> **Three scenarios:**
> Our work is orthogonal to the three scenarios (Hsu et al (2018) and van de Ven & Tolias (2019)) as all rules can be applied regardless of which scenarios the experimenter chose.
> The scenario can be incremental-class learning but this does not imply classes arrive in perfect sequential order (an implicit assumption many authors made).
> Another weakness of the three scenarios is that they only apply in discrete task boundaries.
> To quote van de Ven & Tolias (2019):
> > "If there are no such boundaries between tasks—for
> example because transitions between tasks are gradual or continuous—the scenarios we describe here
> no longer apply, and the continual learning problem becomes less structured and potentially a lot
> harder."
>
> We think this is fundamentally flawed as gradual or continuous is usually defined arbitrarily.
> With clearly defined tasks, we can have tasks alternating frequently.
> Eg considering task 1(1000examples): classify A or B, task 2(1000examples): classify C or D, task 3(1000examples): classify E or F.
> Instead of the typical case of 1000 examples of task 1 then task 2 then task 3, we can have a task change every 10 examples.
> We think this is still a discrete change within the definition.
> Our work also considers continuous scenarios which do not fit in previous work's framework.
>
> **The main result of Hsu et al (2018):**
>
> The main experimental result of Hsu shows that strong baseline methods are competitive.
> In particular, their baseline is replay/rehearsal based.
> This is an application of rule 3 of including simple baseline methods.
> However, our rule focuses on simple methods conforming to experimental settings such as naive multi-model.
> Since, if there is a memory constraint then replay/rehearsal based is not appropriate.
> Furthermore, our work also provides five other rules for robust experimentation and additional analyses on experimental protocol.

---

> > ### Author Response · Authors · 2021-11-22
> > **Cont. of Response to rfDE**
> >
> > **Key result of van de Ven & Tolias (2019):**
> > The key result from van de Ven & Tolias (2019) shows performance deterioration when comparing TIL vs CIL.
> > Our work differs in expanding on the possible types of restriction of the output with a different type of task oracles.
> > In section 4.4 paragraph starting 'For evaluate es', we pointed out that this is the same as multi-head setting and how es-ed and 'td' is a slight variation and perform worse.
> > Td task identifier setting is when we still have a perfect task-oracle like the typical setting except the task definition is different to the data we receive.
> > Eg, the data sequence might be
> > * Chunk1=class{ A,B,C} and
> > * Chunk2=class{D,E,F}.
> >
> > Traditionally, one will assume task-oracle to give
> > * task 1=class{A,B,C} and
> > * task 2=class{D,E,F}.
> >
> > Td might give
> > * task 1=class{B,C,D} and
> > * task 2=class{A,E,F}.
> >
> > This is an important result showing extra assumptions researchers made.
> >
> > We hope to convince the community that it is important to separate task oracle (identifier) from the sequence of data received.
> > As we may know what kind of task we might encounter (ie have task oracle) but can not know when we encountered them.
> > Hence, the rules are designed to cover different possible scenarios which practical applications may encounter.
> > To illustrate, in section 4.4, we mentioned,
> > >"We show that most task oracle based methods
> > not only assume a perfect task identifier which maps examples to the task perfectly but there is also an
> > additional assumption that the group of classes perfectly aligns with the incoming data."
> >
> > We think this is an interesting and novel result as despite having a perfect task-oracle available for multi-head restriction but if the oracle does not match the income data sequence the performance gain is much less prominent.
> > Also, we show that methods with task labels supply for applying their continual learning approach but not for multi-head restriction do not enjoy the large performance gain.
> > Further demonstrating, task oracle based protocols are mostly benefiting from multi-head aligning with data sequence rather than simply being multi-headed or the methods themselves.
> > Having multi-head or task-oracle is not enough but they must match the data sequence encountered.
> > If one can assume data sequence then they could shuffle the data.
> > Which is a strong assumption to make.
> > These results motivate researchers to either remove the assumptions or provide other fairer baselines utilising the assumptions.
> >
> > We thanks the reviewer for the minor edits.

---

> > > ### Comment · Reviewer_rfDE · 2021-11-22
> > > **Response to response**
> > >
> > > Thanks to the authors for their carefully considered response. Their response has brought out the differences between their work and that of van de Ven & Tolias (2019) and Hsu et al (2018). I have increased my score accordingly.
> > >
> > > Apologies for my enigmatic comment about the above two works being arXiv only publications. The authors ask me "We are unsure of the meaning implied. Is it that arXiv results can be disregarded as they are not formally published or if arXiv results are important." No, I certainly didn't mean either of those things. Indeed, I agree that those two works have been important and influential. What I meant was detailed scholarly work like this (and like yours) - valuable as it is - is not the type of work that usually appears in major conferences. Perhaps this should change, but that's a meta-level decision for the field (or at least for the AC)

---

### Official Review · Reviewer_tqw3 · 2021-10-29

**Correctness:** 3
**Technical Novelty And Significance:** 2
**Empirical Novelty And Significance:** 2
**Recommendation:** 5
**Confidence:** 3

**Main Review:**

The paper addresses an important problem in continual learning and the introduced rules are clear and intuitive. Authors also have done a great job capturing various types of data sequences and task identifiers. The following are the ares of improvement:
- Discussions in the paper are mostly dry and are raised in the vacuum. Providing examples would have been helpful for the following sample cases:
-- P4: "However, in many applications, tasks are either not known". what kind of applications?
-- P6: What are some examples of tasks when "unrestricted scenario" becomes important?
- Naive exploration of the hyperspace: The paper identifies a few experimental design dimensions and multiple variations for each, however, it does not provide much suggestion about how these dimensions and their different variations should explored or which combination should be used to evaluate different continual learning methods. Without shedding light into this matter, the paper is regarded more as a study providing evidence for the role of experimental design on continual learning accuracy which is not a surprise.
- Limited hindsight analysis of the existing work: Section 4 provides experimental results for a few scenarios including unrestricted and reappearing classes. However, the flow of the experiments is not well justified and only limited hindsight analysis is provided (e.g. P7: In the unrestricted setting for splitMNIST, most regularisation results are around 20% which is a similar founding as Farqhuar Farquhar & Gal (2018) as their single head setting) where the previous findings seem to be aligned with the experimental results reported in the paper. It would have been more useful to report cases of misalignment between the experimental results reported in the paper and the existing literature to highlight the importance of the experimental settings on the  previous bodies of works' conclusions.
- Minor comments
-- P2: ensure that the method => ensures that the method
-- Tables in section 4 are not sell explanatory. It's not clear what are the numbers reported in the table and one need to read through the text to figure what they actually represent.

**Summary Of The Paper:**

The paper proposes a experimental design unification scheme for continual learning methods by introducing six rules and highlight the implications of the experimental settings on the accuracy of different methods. The effect of four types of sequences of incoming data and three types of task identifier is empirically investigated on the accuracy of different continual learning methods.

**Summary Of The Review:**

Overall my recommendation is to reject the submission in it's current form, however, if the authors can address the comments I am fine with accepting it as a marginally above the threshold submission. The rationale is that:
1- Robustness of experimental settings for continual learning is important and the paper takes steps towards formalizing and unifying it.
2- Reported experimental results and the discussions (although limited) are still useful for the community.

---

> ### Author Response · Authors · 2021-11-22
> **Response to tqw3**
>
> We thank the reviewer for their suggestions. Following is our response to the reviewer:
>
> **Q1 "However, in many applications, tasks are either not known". what kind of applications?:**
>
> We think the abundant existing works on the task agnostic methods justifies this point.
> The other reason is the sequence of data should not be assumed even when we have perfect task oracles.
> On a more philosophical aspect, task definition is ambiguous.
> Hence, if one can define tasks then they usually have a lot of information about the dataset.
> For example, current work in CL will arbitrary slice up the CIFAR dataset into tasks.
>
> **Q2 What are some examples of tasks when "unrestricted scenario" becomes important?:**
>
> An ambiguity of the class incremental learning (CIL) scenario is whether or not the underlying methods can assume a particular data sequence.
> We hope to convince the community with our rules and settings that the two should be untangled and assumptions made explicit.
> CIL assumes discretely encountering tasks and task label is not given in test time.
> However, it is unclear if a task label is given during training.
> Our setting aims to remove the ambiguity considering data sequence, task-oracle sequence and, availability of task-oracle separately.
> The unrestricted scenario only assumes a task-identifier is available for training but not testing.
> Note it does not assume a perfect task identifier hence when training it only keep track of what classes has already been seen before and predict from already seen classes.
> We do not assume sequential data sequence or task-oracle sequence.
> Hence, multiple sequences are tested and shown in the results.
> For Q4, CIL is equivalent to sp=5 identifier in splitMNIST and sp=10 for splitCIFAR.
>
> **Q3 Naive exploration of the hyperspace:**
>
> We think the exploration of hyperspace is important and interesting due to a large number of combinations.
> Our rules such as rule 1 and 5 are suggestions into the possible hyperspace by looking into setting such as the extremes.
> We think further work on this topic will be useful for CL research but is beyond scope of our work and an important problem on its own.
>
> **Q4 Limited hindsight analysis of the existing work:**
>
> The purpose of our work is to point out the large variation of results rather than any specific performance value.
> The same single head setting (sp=5) in unrestricted still holds for splitMNIST as shown in Table 7. (This result should match Hsu's work)
> Looking across columns we see some variations.
> Interestingly having weaker task identifiers such as Sp=2 or even random still retain or outperform sp=5 (perfect match with data sequence) for the case of iCaRL and regularisation.
> This is further reinforced Q1 showing the handpicked task oracle may not be the best one and is an assumption made.
> Naive multi-model (which does not completely adhere to the setting) do maintain the best performance for sp=5 whereas others are least perfectly matching as reflected in their random counterpart.
> We have added additional explanations into the appendix section D2.
>
> We focused on CIFAR100 as it is a more difficult dataset.
> Hsu and van de Ven & Tolias only focused on MNIST.
> We think we have compared the two main results in section 4.3 and 4.4 with section 4.2 (existing work).
> For example, in sec4.3 we mentioned "The results shown in Table 3 are similar to the original splitCIFAR but with slightly lower performance.
> This is likely due to the training data being further spread out hence more forgetting will occur."
> In section 4.4 paragraph starting 'For evaluate es', we pointed out that this is the same as multi-head setting and how es-ed and td is a slight variation and perform worse.
> Td task identifier setting is when we still have a perfect task-oracle like the typical setting except the task definition is different to the data we receive.
>
> Eg, the data sequence might be
> * Chunk1=class{ A,B,C} and
> * Chunk2=class{D,E,F}.
>
> Traditionally, one will assume task-oracle to give
> * task 1=class{A,B,C} and
> * task 2=class{D,E,F}.
>
> Td might give
> * task 1=class{B,C,D} and
> * task 2=class{A,E,F}.
>
> This is an important result showing extra assumptions researchers made.
>
> We thank the reviewer for other minor suggestions and have made edits where possible.

---

### Official Review · Reviewer_vS9u · 2021-10-31

**Correctness:** 2
**Technical Novelty And Significance:** 1
**Empirical Novelty And Significance:** 2
**Recommendation:** 1
**Confidence:** 2

**Main Review:**

Personally, I think the paper makes limited contributions to the evaluation of continual learning. I am very surprised that the paper didn’t mention different settings of continual learning and seems to mix up the different settings. There are three main continual learning setups, class incremental learning, task incremental learning, and domain incremental learning. Different settings require different evaluation methods. Without discussing the settings, the value of the paper drops dramatically.

I think the proposed rules have some merits but quite limited. Most rules are related to the sequence of tasks, e.g., rule 1 and rule 5. But different sequences of tasks have been considered in testing by several existing papers, e.g., Ke et al. Continual Learning of a Mixed Sequence of Similar and Dissimilar Tasks. NeurIPS-2020.

I also have issues with some aspects of some rules. For example, rule 2 says that providing a random guess classifier conforming to the experimental protocol is important to demonstrate the performance improvement attributable to the method. Personally, I do not see the value of this with the current state of the art. If a system does not do better than random guessing, it cannot beat any baselines. I also do not see that separating a problem into different chunks or tasks has anything wrong because many practical problems are like that. I agree that some mixing of tasks will be useful in evaluation, but that is similar to a replay-based method.

Rule 3 says testing simple methods, but again several papers have already done so, e.g., multitask learning, building a separate model for each task, and training all tasks incrementally without any forgetting prevention mechanism.

Regarding task identifiers, I do not see any issue of providing them in training. But for testing, it depends on what setting of continual learning is used. For task incremental learning, task identifiers should be provided because they represent completely independent tasks, e.g., given the same image of a dog, one task may want to classify whether it is an animal or not, but another task may want to classify it as a specific breed of dog. Without the task identifier provided, how to classify it? For class and domain incremental learning, task identifiers should not be provided in testing.



**Summary Of The Paper:**

The paper proposed 6 rules for robust continual learning evaluation and conducted some experiments.

**Summary Of The Review:**

I think the paper made limited contributions to the evaluation of robust continual learning. It doesn't contain much more than what are already known to the research community. This reviewer also doesn't agree with some points.

---

> ### Author Response · Authors · 2021-11-22
> **Response to vS9u**
>
> We thank the reviewer for their comments. Following is our response:
>
> A key contribution and message of our paper is to give a useful set of rules that all should be followed when experimenting in continual learning (or more general ML settings).
> Continual learning methods' abilities are hard to grasp due to the vast and messy settings available.
> Despite some rules being followed in some work, most works still lack adherence to multiple rules.
> The rules also is for moving towards a more online and sequential base framework by reducing the bias of experimental design orientating with task.
>
> We hope to convince the community that it is important to separate task oracle (identifier) from the sequence of data received.
> As we may know what kind of task we might encounter (have task oracle) but can not know when we encountered them.
> Hence, the rules are designed to cover different possible scenarios which practical applications may encounter.
> To illustrate, in section 4.4, we mentioned, "We show that most task oracle based methods
> not only assume a perfect task identifier which maps examples to the task perfectly but there is also an
> additional assumption that the group of classes perfectly aligns with the incoming data."
> We think this is an interesting and novel result as despite having a perfect task-oracle available for multi-head restriction but if the oracle does not match the income data sequence the performance gain is much less prominent.
> Also, we show that methods with task labels supply for applying their continual learning approach but not for multi-head restriction to not enjoy the large performance gain.
> Further demonstrating, task oracle based protocols are mostly benefiting from multi-head aligning with data sequence rather than simply being multi-headed or the methods themselves.
> Having multi-head or task-oracle is not enough but they must match the data sequence encountered.
> If one can assume data sequence then they could shuffle the data.
> Which is a strong assumption to make.
> These results motivate researchers to either remove the assumptions or provide other fairer baselines utilising the assumptions.
>
> **Q1 Three scenarios:**
> Our work is orthogonal to the three scenarios as all rules can be applied regardless of which scenarios the experimenter chose.
> The discussion has been omitted due to the page limit and the orthogonality.
> The scenario can be incremental-class learning but this does not imply classes arrive in perfect sequential order (an implicit assumption many authors made).
> Another weakness of the three scenarios is that they only apply in discrete task boundaries.
> To quote van de Ven & Tolias (2019):
> "If there are no such boundaries between tasks—for
> example because transitions between tasks are gradual or continuous—the scenarios we describe here
> no longer apply, and the continual learning problem becomes less structured and potentially a lot
> harder."
> We think this is fundamentally flawed as gradual or continuous is usually defined arbitrarily.
> With clearly defined tasks, we can have tasks alternating frequently.
> Eg considering task 1 (1000examples): classify A or B, task 2(1000examples): classify C or D, task 3(1000examples): classify E or F.
> Instead of the typical case of 1000 examples of task 1 then task 2 then task 3, we can have a task change every 10 examples.
> We think this is still a discrete change within the definition.
>
> **Q2 Sequence of tasks and dissimilar tasks:**
> Rule 1 is not just for the sequence of tasks and is meant to act more general.
> Initially, rule 5 was also designed to be more general such as considered choices like the batch size.
> However, we decided to focus on the sequence as this seems to be the most dominant effect on experimental results.
> As mentioned, we wish to convince the community that the rules should be applied to all work rather than that it has already been tested in one work.
> Since every work is likely to have slightly different protocols, implementations and choices that may affect the results.
> We also think that our rules apply to Ke et al.'s work as their work still assume sequential data.
> Ke et al do not provide a formal definition of what constitutes similar or dissimilar tasks.
> Hence, further assumptions are built into their protocol.
> We will give a detailed example of our rules for the Ke et al. paper at the end of this response.
>
> **Q3 rule 3 simple method:**
> Again our rule is suggesting all work to repeat these results for their method and protocol.
> Performance is difficult to compare due to the volatile effect of different experimental design.
> The simple method is to adhere to the specific protocol for each specific study.
> Note a substantial amount of recent publication in top conferences that is task-oracle based omit this result.
> Whereas, for the work that includes simple methods many fail to beat the simple baseline.
> Note, for example, Ke et al paper barely outperform this simple method and in numerous settings performing worse.

---

> > ### Author Response · Authors · 2021-11-22
> > **Cont. response to vS9u**
> >
> > **Q4 rule 2 baseline and mixing of task:**
> > Random guessing provides the clearest and direct baseline to comply with the protocol.
> > This gives an empirical measure of the ability the protocol and experimental design is supplying.
> > Separating into tasks or chunks is not a problem.
> > However, here are two assumptions: 1)what will task-oracle tell us about the data? 2) what is the data sequences/composition
> > The problems are entwining the two assumptions into one (so we think we are making a weaker assumption) and not utilising assumptions for baseline comparisons.
> > We also think if you can separate data into chunks you probably have enough control over it to shuffle before you make the chunks.
> > The data is not stored as our experiments are conducted in an online setting.
> > Whereas, replay based method requires extra memory bandwidth.
> > We also think we should avoid mixing methods and protocols.
> > Since you can have mixed sequences of data for the replay based method.
> > Similar, replay based method results do not translate to other CL methods with the mixed sequence of data.
> > Assuming a specific sequence of data is also dangerous as it can act as a proxy of task change such as switching to a different model.
> > Hence, one should test different sequencing to confirm methods apply more generally.
> >
> > **Q5 Task identifiers in training and testing:**
> > As mentioned in our paper, when researchers think this is a valid assumption to make they should provide sensible baselines for fairer comparison (which many fail to beat).
> > On a more philosophical aspect, task definition is ambiguous.
> > Hence, if one can define tasks then they usually have a lot of information about the dataset.
> > Otherwise, we think it makes sense to try various compositions of task.
> > > "For task incremental learning, task identifiers should be provided because they represent completely independent tasks, e.g., given the same image of a dog, one task may want to classify whether it is an animal or not, but another task may want to classify it as a specific breed of dog."
> >
> > we think this is a good application on when task labels could be supplied.
> > However, despite this motivating application, this is not what most TIL continual learning test but rather on artificial splits of MNIST or CIFAR.
> > Which is a different application and we think should use a different protocol.
> >
> >
> > Example of rules for Ke et al
> > Here we give some suggestions of protocol or choices to follow:
> >
> > 1. Rule 1 Extreme settings:
> > * Test random data (mixture of both datasets fully).
> > * Test where all tasks are similar tasks.
> > * Test where all tasks are dissimilar tasks.
> >
> > 2. Rule 2 Random baseline:
> > Provide random baseline. NCL can potentially act as a proxy but would still be useful to see the raw ability of the protocol.
> > (Perhaps NCL might perform similar or worse)
> >
> > 3. Rule 3 Simple methods:
> > The paper provides ONE model and NCL baseline which is an example of following this rule.
> > Applying validation set for task similarity to ONE model is a possible improvement with the view of adherence to protocol.
> >
> > 4. Rule 4 Recapitulate task:
> > We suggest including experiments that recapitulate previously seen tasks.
> >
> > 5. Rule 5 Inconquential experimental choice:
> > Can try a different size of chunks.
> > It seems all the data is of more number of similar or dissimilar tasks.
> > Should provide results where there is an equal number or more dissimilar tasks.
> >
> > 6. Rule 6 metric should depend only on data:
> > Forward transfer is defined as the test accuracy of each similar task when first learned.
> > This is a violation of the rule as it assumes what sequence of data will arrive.
> > Backward transfer interestingly is defined as accuracy after all training which is just the same as final accuracy. (Evidently, table 2 and table 4 result is the same)
> > Not sure why they give this redundant definition.

---

### Official Review · Reviewer_Ajqh · 2021-11-04

**Correctness:** 3
**Technical Novelty And Significance:** 2
**Empirical Novelty And Significance:** 2
**Recommendation:** 3
**Confidence:** 5

**Main Review:**

**Strengths**
- All rules are sound
- enough baselines are provided

**Weaknesses**
I have some problems with the rules
- Rule 1
    - I think this rule as merit, but the authors should provide some motivation for it
- Rule 2
    - although a good rule, this is already reported in most high-quality research papers, or it is trivial to compute.
- Rule 3
    - also reported in most high-quality research papers
- Rule 4
    - good rule to bring up, although, as the authors pointed out, it has been studied in OSAKA.
- Rule 5
    - same as above
- Rule 6
    - not sure I understood fully this rule. I think the text could be improved

also, related to the experiments and discussion:
- "The availability of task labels for testing simplifies the learning problem significantly in task oracle base methods."
    - this is not a new finding, it's well reported in the literature. Actually, lots of papers report both performance as class-incremental vs task-incremental learning.
- "in OSAKA (Caccia et al., 2020). The work considers task transition to be model by a Markov chain and uses $\alpha$ parameter to decide how likely the next data will be drawn from the current task. Hence, recapitulation occurs when it transitions back to previously visited task. However, OSAKA did not consider an extreme scenario as suggested by rule 1 by considering very low $\alpha$ values where transitions between tasks are prevalent."
    - IIUC, $\alpha$ controls the non-stationarity of the distribution and lowering $\alpha$ brings us closer to a stationary data distribution, which is by definition outside the scope of Continual learning.

**Summary Of The Paper:**

This work proposes 6 rules for continual learning experiments.
Specifically, they propose to
- 1) test for extreme continual learning settings,
- 2) report random guess performance
- 3) evaluate naive/simple methods
- 4) revisit old data
- 5) test for different task orders and
- 6) define performance that derive from the data.

Then,  pan empirical study on different scenarios that emerge from the combinations of those rules is provided.

**Summary Of The Review:**

I think this work as merit and could potentially make a good contribution to the field  if improved.
I don't think it's top-conference material in its current form, however.
I am sorry to the authors, but I do not have more constructive criticism to provide at the moment.

---

> ### Author Response · Authors · 2021-11-22
> **Response to Ajqh**
>
> Thanks to the reviewer for their feedback and comments. Following is our response to the reviewer:
>
> ## Scattered discourse
> A key contribution and message of our paper is to give a useful set of rules that all should be followed when experimenting in continual learning (or more general ML settings).
> Continual learning methods' abilities are hard to grasp due to the vast and messy settings selected.
> Despite some rules being followed in some work, most works still lack adherence to multiple rules.
> The rules also is for moving towards a more online and sequential base framework by reducing the bias of experimental design orientating with task.
> Applications of the rules are to aid the design and choices of experiments and results for a better understanding of the methods.
> Most continual learning protocol applies to very specific applications eg multi-task learning where there are different tasks for example such as image segmentation and identification.
> In the multi-task learning setting, task label may make a lot of sense as the model need to know which task to perform.
> However, many works will use this protocol for applications (dataset) that do not need this. Eg MNIST or CIFAR.
>
> Rule 2, 3 and 6 are especially useful to give comparisons for understanding the results.
> Another aspect is to reduce assumptions of the knowledge of the task in continual learning.
> We think that continual learning researchers should move towards a more online setting (such as OSAKA) and avoid assuming sequential knowledge of data.
> As in most applications, we cannot enforce the data to receive in certain. (Otherwise, if we did have this ability we could just shuffle the data).
> Hence, Eule 1, 4 and, 5 would be useful to test cases where these assumptions won't hold to give a more general understanding of the methods.
> Finally, we want to note that the rules are not necessarily mutually exclusive.
> Protocols may be inspired by a rule but have the properties of multiple rules.
>
> Now we address the specific concerns of the reviewer:
>
> ## Rule 1
> Researchers should not assume the sequence in which data will arrive as mentioned earlier.
> Suppose we do know the sequence then this usually means we have full control over the dataset and can just shuffle the dataset.
> It is infeasible to cover all settings.
> Hence, to provide a border landscape of the results we propose this rule to cover the extremes.
> For example, extremes in sequential-ness of tasks.
> The most sequential-ness will be the typical setting of continual learning where tasks are disjoint and arrive one after another.
> No sequential-ness is each example is a different task.
>
> Another could be task-oracle abilities.
> On one hand, we can have a perfect task-oracle that always give the correct task.
> On the other, we could have a task-oracle that gives random task labels.
>
> Note rule 1 is related to rule 5 as many extremes can be considered in the composition of the data such as the number of tasks.
> eg even if data arrives sequential (not known to model), the task oracle may treat the entire dataset as one task.
>
> ## Rule 2
> We share a similar view on rule 2 is a useful rule to include and is not difficult.
> However, strikingly many top recent publications still lack this result.
> Hence, these rules are included to remind researchers.
> Another important aspect is the random classifier should adhere to the experimental setting used for testing the main continual learning methods.
> For example, the result is 1.5\% which is different to the intuitive result of 1\% in table 4 and column **td** for random classifier following its setting.
>
> ## Rule 3
> Similar to above, many papers still do not include these results.
> As before, the simple method needs to conform to the experimental setting.
> Since sometimes the given simple method result is on a simpler or different setting which is hard to act as a fair comparison.
> Also, many methods specific for task incremental learning fail to beat this baseline yet is still published in top conferences (as mentioned in our paper some consider this as upper-bound).
>
> ## Rule 4
> The reviewer mentioned this is revisiting old data but it is more general on revisiting old tasks but possible with new data.
> The results we provide are visiting previous tasks but with new data.
> For example, a task which consists of data from cats, dogs, horses, rabbits and, tigers are seen then after many iterations this task is revisited (with new unseen data of classes cat, dog, etc).

---

> > ### Author Response · Authors · 2021-11-22
> > **Cont. of response**
> >
> > As before, we think all of the rules should be followed rather than the existence of work following the rules.
> > OSAKA is the only work (to our best knowledge) that attempted this.
> > Different to OSAKA, our experiments enforces strict repeat that is guaranteed to revisit task for sequential task setting.
> > We also provide results of this setting for various methods, task agnostic and task-oracle base methods (in section 4.3 of our paper).
> > Note this setting can still be important for task-oracle base methods that use task labels to detect task changes to activate continual learning such as Synaptic Intelligence and EWC.
> >
> > ## Rule 5
> > We wish the rule is applied for all work rather than just one work.
> > OSAKA did have some attempts in applying this rule such as trying different $\alpha$ values to give different compositions.
> > However, rule 5 is more general and OSAKA is an example of the application of the rule.
> > Due to the ambiguity of what constitutes tasks, different compositions of tasks should be tested.
> > For example, earlier work often uses the numerical sequence of ground truth as the task.
> > OSAKA is also a stochastic application of the rule and our work cover other deterministic (especially useful for task-oracle base methods) settings.
> > For example, we provide additional settings like different task identifier (such as dominant task identifier) or data sequence which is not seen (to our best knowledge) in existing continual learning work on experiment protocols such as Hsu et al (2018) and van de Ven & Tolias (2019).
> >
> > ## Rule 6
> > The main purpose of rule 6 is to remind researchers to not assume sequence in the perspective for evaluation metric.
> > For example, in (Lopez-Paz et al., 2017) backwards transfer (BWT) is defined as $\dfrac{1}{T-1}\sum_i R_{T,i}-R_{i,i}$.
> > If there is even a tiny bit of data from the previous tasks reappearing then $R_{T,i}\sim R_{i,i}$ then BWT is about 0.
> > Hence, this metric is designed with too much assumption of the data sequence.
> > Rule 6 advise the metric to only depend on the data itself and not how it might be presented as this should not be known in the continual learning setting.
> >
> > "The availability of task labels for testing simplifies the learning problem significantly in task oracle based methods.
> > this is not a new finding, it's well reported in the literature. Actually, lots of papers report both performance as class-incremental vs task-incremental learning.":
> > We agree this point is not new.
> > However, we chose to re-emphasise it because we think many task-oracle based CL methods should move away from task-oracle (as they usually do not need to rely on it).
> > Many works in recent top conferences (NeurIPS/ICML/ICLR) still focus on task-oracle based approach and fail to beat naive multi-model baselines.
> > Furthermore, our novel contribution on this aspect is on the new analysis of the different types of task identifiers which allow the performance boost.
> > In section 4.4, we mentioned, "We show that most task oracle based methods
> > not only assume a perfect task identifier which maps examples to the task perfectly but there is also an
> > additional assumption that the group of classes perfectly aligns with the incoming data."
> > We think this is an interesting and novel result as despite having a perfect task-oracle available for multi-head restriction but if the oracle does not match the income data sequence the performance gain is much less prominent.
> > Also, we show that methods with task labels supply for applying their continual learning approach but not for multi-head restriction do not enjoy the large performance gain.
> > Further demonstrating, task oracle based protocols are mostly benefiting from multi-head aligning with data sequence rather than simply being multi-headed or the methods themselves.
> >
> > "IIUC,  controls the non-stationarity of the distribution and lowering  brings us closer to a stationary data distribution, which is by definition outside the scope of Continual learning":
> > As mentioned before, we should not assume the type of sequence we are given (especially in the online setting) hence we might encounter an environment with rapidly changing tasks (low $\alpha$).
> > Note the definition of tasks is still the same as less rapid settings (high $\alpha$) but task changes occur more frequently.
> > We noticed in the OSAKA paper, the result seem to be deteriorating with a lower value ($\alpha=0.9$).
> > Hence, we think this is a good example of where the application of our rule (rule 1 extreme setting of $\alpha=0$) will be useful to give a more robust understanding of the method.
> > As our rules are trying to be simple to perform while providing as many useful and important results we think running such an experiment is fairly easy and a fair result to provide.
> > Another benefit of this result is that it serves as a potential insanity check for the method in a relative non-convoluted setting and detects computational concerns for optimisation that is not general.

---

> > > ### Comment · Reviewer_Ajqh · 2021-11-25
> > > **thanks for the review**
> > >
> > > Thanks for taking the time to write a proper rebuttal.
> > > I tend to agree with most of the points and I think that continual-learning papers attempting to change the status-quo for the better are important.
> > > I, however, still think the contribution is not important enough in its current state to merit a ICLR publication.
> > > I suggest the authors submit to a less competitive tier of ML conferences, or work on the empirical study and try to come up with more new insights.

---

### Decision · Program_Chairs · 2022-01-20

**Decision:**

Reject

**Comment:**

This paper is a scholarly examination for how to conduct continual learning evaluations, proposing six rules that in large part synthesize work from other papers. While there is certainly scholarly benefit to such an exploration, all reviewers believe that the contribution is not substantial enough in its current form to warrant acceptance. It is certainly true that not all continual learning papers follow all of the guidelines/rules for evaluation, and consequently, papers such as this are useful to improve the scientific process.  However, the contribution needs to be substantially deepened, including more extensive and in-depth experiments with novel insights as described in the reviews, before the paper is ready for publication.